# Electronic Nose Sensor Drift Affects Diagnostic Reliability and Accuracy of Disease-Specific Algorithms

**DOI:** 10.3390/s22239246

**Published:** 2022-11-28

**Authors:** Sofie Bosch, Renée X. de Menezes, Suzanne Pees, Dion J. Wintjens, Margien Seinen, Gerd Bouma, Johan Kuyvenhoven, Pieter C. F. Stokkers, Tim G. J. de Meij, Nanne K. H. de Boer

**Affiliations:** 1Department of Gastroenterology and Hepatology, AG&M Research Institute, Amsterdam UMC, Vrije Universiteit Amsterdam, 1081 Amsterdam, The Netherlands; 2Department of Epidemiology and Biostatistics, Vrije Universiteit Amsterdam, 1081 Amsterdam, The Netherlands; 3Biostatistics Unit, Netherlands Cancer Institute, 1066 Amsterdam, The Netherlands; 4Department of Gastroenterology and Hepatology, Maastricht University Medical Centre (MUMC+), 6229 Maastricht, The Netherlands; 5Department of Gastroenterology and Hepatology, OLVG West, 1061 Amsterdam, The Netherlands; 6Department of Gastroenterology and Hepatology, Spaarne Gasthuis Hospital, 2134 Hoofddorp, The Netherlands; 7Department of Pediatric Gastroenterology, UMC, Vrije Universiteit Amsterdam, 1081 Amsterdam, The Netherlands

**Keywords:** electronic nose, volatile organic compounds, sensor drift, inflammatory bowel disease, diagnostic

## Abstract

Sensor drift is a well-known disadvantage of electronic nose (eNose) technology and may affect the accuracy of diagnostic algorithms. Correction for this phenomenon is not routinely performed. The aim of this study was to investigate the influence of eNose sensor drift on the development of a disease-specific algorithm in a real-life cohort of inflammatory bowel disease patients (IBD). In this multi-center cohort, patients undergoing colonoscopy collected a fecal sample prior to bowel lavage. Mucosal disease activity was assessed based on endoscopy. Controls underwent colonoscopy for various reasons and had no endoscopic abnormalities. Fecal eNose profiles were measured using Cyranose 320^®^. Fecal samples of 63 IBD patients and 63 controls were measured on four subsequent days. Sensor data displayed associations with date of measurement, which was reproducible across all samples irrespective of disease state, disease activity state, disease localization and diet of participants. Based on logistic regression, corrections for sensor drift improved accuracy to differentiate between IBD patients and controls based on the significant differences of six sensors (*p* = 0.004; *p* < 0.001; *p* = 0.001; *p* = 0.028; *p* < 0.001 and *p* = 0.005) with an accuracy of 0.68. In this clinical study, short-term sensor drift affected fecal eNose profiles more profoundly than clinical features. These outcomes emphasize the importance of sensor drift correction to improve reliability and repeatability, both within and across eNose studies.

## 1. Introduction

Electronic nose (eNose) technology is a platform in which arrays of cross-sensitive chemical sensors are combined with pattern-based algorithms to identify gaseous mixtures [1]. These mixtures consist of volatile organic compounds (VOCs), which, when measured in (the headspace of) bodily excrements, represent end-products of human metabolism [2,3]. The VOC profile as measured by eNose is a black box in terms of metabolite composition, but its features provide a particularly suitable basis to develop diagnostic pattern-based algorithms. The advantages of this technique include fast, easy-to-perform and low-cost measurements, which underline its usability for clinical practice [4].

A well-known limitation is that sensors are prone to gradual deviations of their VOC-binding characteristics due to differences in temperature and air humidity leading to temporal sensor drift [5]. This is thought to decrease compatibility between sensor output and the developed algorithm in a later stage, which may affect accuracy during clinical studies and hampers both the comparison and validation of results when no precautions are taken [6]. Multiple studies have focused on sensor optimization to prevent drift; however, more advanced sensors are often accompanied with higher costs and more labor-intensive measurements [7,8]. Correction methods are another option to avoid bias caused by sensor drift [9,10,11]. For example, the recalibration of all sensors may be performed using daily study measurements for the direct correction of gas-sensor response. These methods are usually focused on composing drift in ‘baseline samples’ and extracting it from the sensor response. The most commonly proposed methods are Principal Component Analysis (PCA) or Common PCA (CPCA), Independent Component Analysis (ICA) and Discrete Wavelet Transform (DWT) [12,13,14,15,16,17]. This is, however, time-consuming, and a large batch of ‘baseline’ samples are needed, which are often of limited availability. Another method by which to correct for drift is by identifying components unrelated to classification, including proposed methods such as Orthogonical Signal Correction (OSC), Linear Discriminant Correction (LDA) and Partial Least Squares (PLS) [18,19,20,21]. A third way to correct for sensor drift is more universally applicable and involves machine learning such as Kernel Transformation (DCKT), Self-Organizing Maps (SOMs), Adaptive Resonance Theory (ART), and Deep-learning Neural Networks [9,11,22,23,24,25,26,27]. The choice of the optimal fitting correction method is dependent on both the characteristics of the data and the type of sensors used.

Electronic nose technology has been presented as a promising non-invasive tool to detect various diseases (e.g., lung cancer, colorectal cancer, celiac disease) [28,29,30]. Nevertheless, the use of correction methods for sensor drift is not yet standardized in clinical studies. Short-term sensor drift may affect the accuracy of algorithms as early as during data training. As this drift occurs within the ‘black box’, its possible effect modification may go unnoticed. The aim of this study was to investigate the influence of eNose sensor drift on the development of a disease-specific diagnostic algorithm in a real-life cohort. In addition, we proposed a method to correct for this drift. This is illustrated by means of a study on the fecal VOC profiles of patients with inflammatory bowel disease (IBD), as multiple previous studies have identified the eNose-captured VOC profile as a potential diagnostic biomarker [31,32,33,34,35].

## 2. Materials and Methods

### 2.1. Study Design

This multi-center cohort study was performed between April 2016 and May 2018 at the Department of Gastroenterology and Hepatology of one tertiary hospital (Amsterdam UMC, location VUmc) and two district hospitals (Spaarne Gasthuis, Haarlem and Hoofddorp and OLVG West, Amsterdam), all located in the Netherlands. Approval of this study was provided by the Medical Ethical Review Committee (METc) of the Amsterdam UMC under file number 2016.135 and by the local medical ethical committees of the participating centers. All participants provided written informed consent before inclusion in this study.

### 2.2. Participants

#### 2.2.1. Inflammatory Bowel Disease

Patients with an established diagnosis of IBD, aged 18 years and above, and with a scheduled endoscopy were asked to participate in this study irrespective of their endoscopy indication. Patients were excluded in the case of diagnosis with IBD-unclassified (IBD-U) and co-existing abnormalities during endoscopy (e.g., colorectal cancer, polyps, diverticulitis) with the exception of hemorrhoids and diverticula. Other exclusion criteria were history of bowel cancer, failure to perform a complete colonoscopy (independent of the reason), absence of sufficient endoscopy images to evaluate activity of IBD or insufficient fecal sample mass collected to perform a VOC analysis.

#### 2.2.2. Controls

All patients aged 18 years and above with a scheduled colonoscopy at the Amsterdam UMC, OLVG West or Spaarne Gasthuis facilities were asked to participate in this study regardless of the indication for endoscopy. Patients were eligible to participate in this study if no mucosal abnormalities were observed during colonoscopy (with the exception of hemorrhoids and diverticula). In cases where mucosal biopsies were taken, histologic reports were evaluated and subjects were excluded if histologic abnormalities were described by the pathologist. Other exclusion criteria were a history of bowel diseases (except appendicitis), failure to perform a complete colonoscopy (independent of the reason) and/or the collection of insufficient fecal sample mass to perform the VOC analysis.

### 2.3. Data Collection

#### 2.3.1. Sample Collection

All participants collected a fecal sample in a stool container (Stuhlgefäβ 10 mL, Frickenhausen, Germany) prior to colonic lavage and stored these samples in their freezer at home within one hour following collection. On the day of their colonoscopy, participants transported the sample to the hospital in a cooled condition using ice packs. Upon arrival at the hospital, fecal samples were directly stored in the freezer at −24 °C.

#### 2.3.2. Assessment of Variables

All participants completed a questionnaire with items on age, gender, body-mass index (BMI), smoking habits, dietary habits, medical history, comorbidity and use of medication. The IBD patients were matched to controls in a 1:1 ratio, based on age, gender and smoking status as we previously observed that these variables influence fecal VOC profiles [36]. In addition, they were matched according to hospital of inclusion to avoid any possible effects of sample storage on VOC profiles. From the IBD patients, additional disease-specific characteristics were obtained to rule out influence of these variables on sensor output. Data on disease localization and behavior based on the Montreal scores were extracted from the electronic patient records [37]. Information on disease activity at sample collection was obtained by asking two independent gastroenterology consultants to score the endoscopic images. To avoid inter-individual variation in scoring, one of the specialists was present for every assessment. All IBD endoscopy procedures were scored according to the Van der Heide classification, a generic grading system focusing on a broad range of mucosal characteristics [38]. In addition, the mucosal appearance of ulcerative colitis (UC) patients was scored using the Mayo classification (remission defined as 0, mild to severe activity scored as ≥1) [39]. Rutgeerts’ Postoperative Endoscopic Index with scores between i0 (no lesions) and i4 (diffuse inflammation) was used to describe Crohn’s disease (CD) patients with an ileocolic resection [40]. Remission was defined as a Rutgeerts classification of ≤i1 and active disease as ≥i2. The remaining CD patients were categorized into remission and active disease categories based on the opinion of the two consultants.

### 2.4. Sample Preparation

From each original collected fecal sample, 500 mg (with a deviation of 5%) was weighted on a calibrated scale (Mettler Toledo, AT 261 Delta Range, Columbus, OH, USA). This sample amount was chosen based on results of a previous sampling method study for VOC-pattern recognition using field asymmetric ion mobility spectrometry (FAIMS), showing that samples of 500 mg provide an optimum ratio of VOCs to the headspace to distinguish the scent profiles of IBD patients [41]. The subsample was then transferred into a labelled 3 mL sealed vacutainer (BD vacutainer, Franklin Lakes, NJ, USA) and stored once again in a −24 °C freezer. Fecal samples were thawed to room temperature (18 °C) 30 min prior to analysis to allow VOCs to fill the headspace.

### 2.5. Electronic Nose Device

Fecal VOCs were analyzed using an eNose device (Cyranose 320^®^, Smiths Detections, Pasadena, CA, USA). This apparatus operates on an array of 32 nanocomposite sensors. Each sensor consists of carbon nanotubes, conducting polymers and metallic and non-metallic nanoparticles that interact specifically with VOCs in the sample. Interaction between specific sensors and VOCs results in unique changes of electrical resistance, which are calculated per sensor using the following formula: $$\frac{\Delta R}{R_{0}}$$
where ΔR denotes the maximum resistance change of the sensor during each measurement and R_0_ is the baseline resistance before the start of the measurement. As there are 32 sensors in this particular eNose, this leads to 32 unique signals per sample, combined as the so-called ‘smell-print’. As shown in this formula, the baseline resistance has a major influence on sensor output, meaning that temporal sensor drift during sample measurements has an influence on the ‘smell-prints’.

### 2.6. Fecal Volatile Organic Compound Analysis

The VOC measurements were performed on four subsequent days. All samples were measured in random order by a blinded analyst. Analyses were performed using a previously validated standard operating procedure [42]. In Appendix A, a schematic setup of eNose measurements is presented. In short, the vacutainer is connected to the air-tight loop system of the Cyranose 320^®^ tool by piercing two needles (BD Blunt fill needle 1.2 × 40 mm, Franklin Lakes, NJ, USA) into the cap of the vacutainer. These needles are attached to two oxygen hoses (Argyle Covidien tube 3 mm, Mansfield, MA, USA), which are connected to the snout and outlet of the Cyranose 320^®^ tool by three-way valves (stopcock connecta plus, McFarlane Medical, Melbourne, Australia). The valves provide control of the airflow direction. A water filter (Nalgene 25 mm Syringe Filter, Thermo Scientific, Waltham, MA, USA) is connected between the needle and the oxygen hose to prevent condensation in the system and thus contamination. Before the actual VOC analysis is performed, sensors are purged for 90 s with filtered ambient air (VOC filter A1, North Safety, Middelburg, The Netherlands) to detach residual VOCs from the eNose sensors. Then, a baseline measurement using ambient filtered air takes place for 30 s. Subsequently, the actual measurement is performed for 60 s, followed by purging of the inlet for 15 s and purging of the sensors for 90 s, to remove the VOCs from the sensors for the next analysis. After each sample analysis, the needles, tubes, connectors and three-way valves are replaced to prevent contamination.

### 2.7. Statistical Analyses

All raw sensor output data, including our R script and output, are openly available at the following link: https://doi.org/10.6084/m9.figshare.21547794.v1 (accessed on 18 November 2022). Demographic data were described and compared using IMB SPSS statistics version 25. Levene’s test was used to assess the assumption of equal variances; subsequently, one-way ANOVA or Kruskal–Wallis tests were used to compute differences in baseline demographics between the control groups and the following two phenotypes of IBD: Crohn’s disease (CD) and ulcerative colitis (UC). Data from the eNose were analyzed using R. Spearman’s correlation was used to assess association between sensor outcomes when exposed to fecal samples. Then, heatmaps of the sensor output were obtained using scaled outcomes per sensor by first computing the mean and standard deviation per sensor across all samples, followed by subtracting each measurement from the mean and dividing it by its standard deviation, so that all sensor measurements are on the same scale across all sensors. Then, the characteristics of samples were added to the heatmap, including the date of the analysis. The eNose data displayed an association with the date of measurement which was reproducible across samples. We proposed the use of a correction factor for the date of sample measurement based on the assumption of Component Correlation (CC), in which it is assumed that all sensors behave in a similar way with respect to drift and the vector of drift [10]. We fitted a regression model based on the control samples to correct for the date of measurement for all samples. Briefly, the mean relative sensor deviation per day was calculated per sensor for all control samples and sensor outcome for cases were corrected based on these data. The residuals of this model applied to the patient data then yield date-corrected measurements, which were used for subsequent analysis. Differences in sensor outcome between groups were then calculated and a logistic regression analysis was performed on the sensors which differed significantly between groups. When significant differences were found, leave-one-out cross-validation was performed using optimum cut-off values, and a prediction model was constructed.

## 3. Results

### 3.1. Baseline Characteristics

Baseline demographics of study participants and endoscopy indication are described in Table 1. A total of 63 IBD patients were included. Of these, 24 were CD patients (19 active disease, 5 remission) and 39 were UC patients (22 active disease, 17 remission). Out of the 227 included controls, a total of 63 were strictly matched to IBD patients based on age, gender, smoking status and hospital of inclusion. Information on Montreal classification, disease activity and medication use of IBD patients is provided in Appendix A.

### 3.2. The Effects of Sensor Drift on Fecal Volatile Organic Compound Profiles

Based on Spearman’s correlation coefficient, it was revealed that sensor outcomes were moderately to highly correlated for all but three sensors, specifically numbers 6, 19 and 24 *(*Figure 1).

A sensor output heatmap is depicted in Figure 2. There was a clear separation between samples measured on the first two dates and samples measured on the second two dates, irrespective of their disease state, disease activity state, disease localization and diet of participants. We observed that all sensor outcomes were on or below their average outcome across samples for the first two days of measurements and on or above for the second two days of measurements. Additional principle component analysis (PCA) charts present the distribution of sensor data based on date of measurement. These PCA charts are included in the R output file, which is openly available at the following link: https://doi.org/10.6084/m9.figshare.21547794.v1 (accessed on 18 November 2022).

As mentioned in the Materials and Methods section, a correction for sensor output was calculated based on control samples using logistic regression. A scaled heatmap after sensor correction was constructed and is depicted in Figure 3. We observed an improved distribution of the measurement date across all samples. In the online available R output file, additional PCA charts present the distribution of sensor output based on date of analysis prior to and after correction for date of measurement (https://doi.org/10.6084/m9.figshare.21547794.v1, accessed on 18 November 2022).

### 3.3. Differentiation between Inflammatory Bowel Disease and Controls

Prior to data correction, no significant differences were seen between fecal VOC profiles of IBD patients and controls. Based on the corrected data, fecal VOC profiles differed between IBD patients and controls based on a total of six sensors (*p* = 0.004; *p* < 0.001; *p* = 0.001; *p* = 0.028; *p* < 0.001 and *p* = 0.005). Sensor specifications with corresponding *p*-values are depicted in Figure 3. Based on the significant sensors, a logistic regression analysis and subsequently leave-one-out cross-validation were performed. The resulting prediction model for IBD patients versus controls is provided in Figure 4. Using an optimal cut-off value of 0.6, the total accuracy was 0.68 with a sensitivity of 0.78 and specificity of 0.59. The positive and negative predictive value (PPV and NPV) were 0.65 and 0.73, respectively. The prediction error distribution was similar for IBD and healthy controls, reflecting the equal group sizes. The Brier score, depicting the mean squared prediction error independent of a cut-off, was 0.2. Box plots of prediction errors are depicted in Appendix A.

## 4. Discussion

In the present study, we have demonstrated that short-term eNose sensor drift affects fecal VOC profiles throughout the measurements of a real-life clinical study and subsequently influences its diagnostic potential.

Though both result in lower diagnostic potential, long-term sensor drift caused by surface aging and sensor poisoning, for example, has been described to have a greater effect on sensor outcomes compared to short-term drift in the available literature [6]. In the current study, however, a worrisome effect of short-term sensor drift was observed. Multiple factors may have contributed to this drift. Even though the standard operating procedure was identical for all measurements, the flow from the airvalve may have varied throughout measurements. This may consequently have led to differences in gas dispersion and total gasflow at specific sensors over time. In addition, when connecting the fecal sample to the eNose setup, exposure of the fecal VOC headspace to the environment cannot be avoided for a brief period of time. Therefore, environmental variations such as ambient air humidity, temperature and contamination with laborant specific odors may have affected the VOC composition when connecting the sample to the eNose setup.

Although sensor drift is a limitation of eNose technology, its high diagnostic accuracy presented in previous studies, combined with its low costs and fast measurements, is appealing for routine testing in clinics [4]. In addition, eNose devices are usually small and may easily be transported from patient to patient, underlining its practicality for in-hospital use. Different approaches for the correction of drift have been proposed; however, this form of correction is not yet routinely used. Most correction methods for sensor drift are reported once and are mostly based on available, public and mostly artificial online environments. The aim of the current study was to present the effects of sensor drift on a real-life cohort.

In the current study, a thorough assessment of patient characteristics was carried out by collecting information on age, gender, BMI, smoking habits, diet, disease localization, behavior and disease activity derived by endoscopy, which is the gold standard. By doing so, we were able to demonstrate that sensor drift has a major influence on ‘smell-prints’, regardless of the participant-specific variables. In addition, a standardized protocol for sample storage conditions and measurements was used, which was based on previous research and minimized variations in sampling conditions [41,42]. Nonetheless, this study also has some limitations. First, patients collected the fecal samples in a non-regulated setting, which means that contamination with VOCs from the environment cannot be ruled out. Second, although we attempted to correct for sensor drift, our current study protocol does not include standardized data collection of ‘blank’ samples containing only filtered air between sets of study samples. Therefore, our correction factors were calculated using the control group in which, although small, differences in fecal VOC profiles are expected. This may have influenced the final corrected outcomes to some extent. Third, it should be noted that, although each sensor of the Cyranose 320 ^©^ tool is thought to have a distinct change in electrical resistance upon contact with a volatile organic compound mixture, most sensor outcomes were correlated in the current study. This suggests that these polymer sensors have a common rather than specific reaction to fecal samples.

For future studies aiming to use eNose technology, we strongly recommend including correction for temporal sensor drift in the standard operating procedure. Researchers should assess the effects of sensor drift on the measurements per study separately, as the extent may vary depending on the eNose used and the moment of measurements. This should be performed in a standardized manner depending on the input characteristics and sensor type. In addition, to avoid inaccurate results with the diagnostic algorithm, we strongly recommend spreading case and control samples evenly throughout the days of measurements.

## 5. Conclusions

In conclusion, short-term eNose sensor drift affects VOC profiles as early as during study measurements, affecting diagnostic accuracy when performing clinical trials. Raising awareness on this topic amongst clinicians is important as correction for this variation improves the reliability and repeatability of eNose study outcomes.

## Figures and Tables

**Figure 1 sensors-22-09246-f001:**
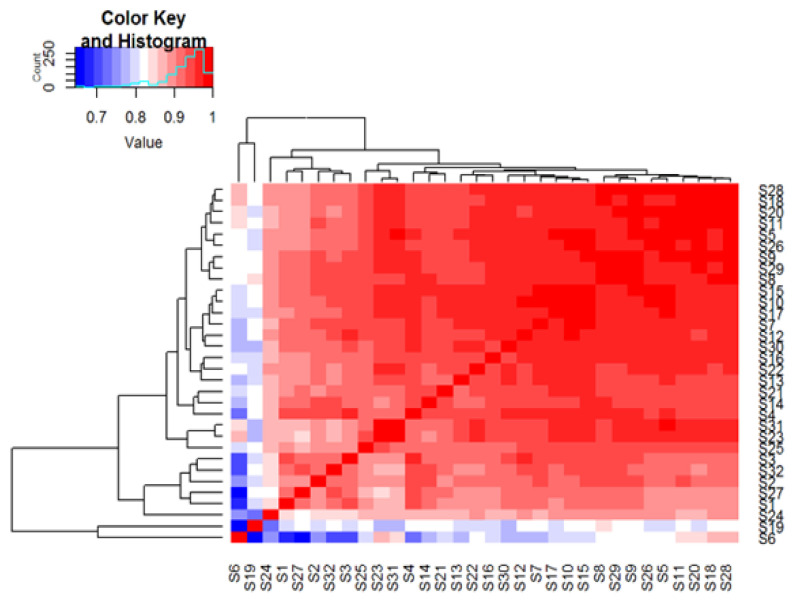
Association between sensors. Spearman correlation of sensors. Sensors are depicted on both the X-axis and Y-axis. A strong positive correlation is seen amongst the majority of sensors.

**Figure 2 sensors-22-09246-f002:**
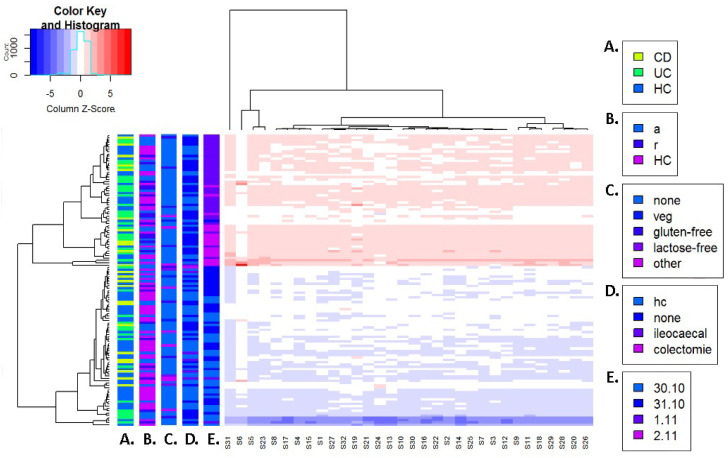
Scaled heatmap of sensor output before correction for date of measurement. The mean and standard deviation per sensor across all samples were calculated. Then, each sample measurement was subtracted from the mean sensor outcome and divided by its standard deviation. This was performed across all sensors and samples so that all sensor measurements were on the same scale. Samples are depicted on the Y-axis, while sensors are depicted on the X-axis. Colored strips on the Y-axis depict the distribution of sample characteristics amongst the samples. (**A**) Distribution of CD, UC and controls. (**B**) Distribution of active disease or remission; control samples are depicted as separate group. (**C**) Distribution of diet including no diet, vegetarian diet, gluten-free diet, lactose-free diet and non-specified diets. (**D**) Distribution of abdominal surgical history except appendectomy, including no abdominal surgery, ileocecal surgery and colectomy; control samples are depicted as separate group. (**E**) Distribution of measurement date.

**Figure 3 sensors-22-09246-f003:**
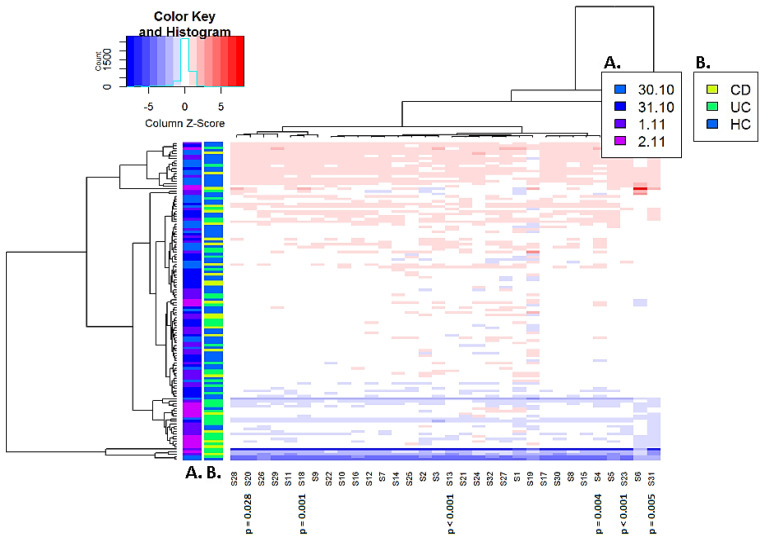
Scaled heatmap of sensor output after correction for date of measurement. Scaled heatmaps of the sensor output. Date of sample measurement was found a confounding factor. Therefore, a correction was applied for date of measurement for all sensor outcomes. Then, the mean and standard deviation per sensor across all samples were calculated. Each sample measurement was subtracted from the mean sensor outcome and divided by its standard deviation. This was across all sensors and samples so that all sensor measurements were on the same scale. Samples are depicted on the Y-axis, while sensors are depicted on the X-axis. Colored strips on the Y-axis depict the distribution of sample characteristics amongst the samples. (**A**) Distribution of measurement date. (**B**) Distribution of CD, UC patients and controls. The six discriminating sensors are indicated by the *p*-values for the discrimination between IBD and controls.

**Figure 4 sensors-22-09246-f004:**
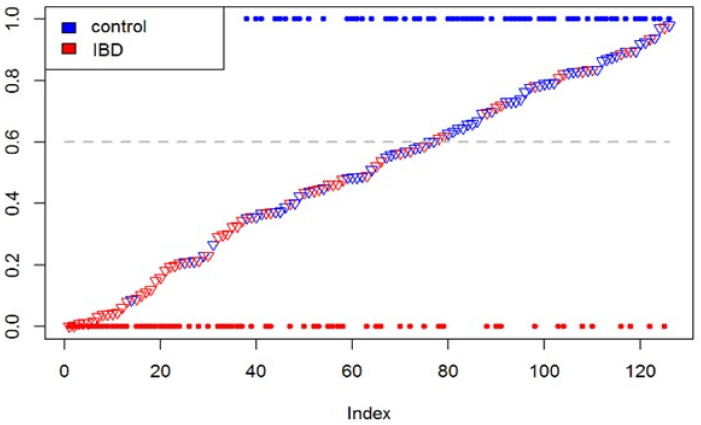
Prediction of inflammatory bowel disease based on leave-one-out cross-validation. Dashed line indicates a cut-off value of 0.6. Y-axis depicts the predicted probabilities of being healthy (so no IBD) for all samples. On the X-axis, the samples are depicted, sorted by predicted probability of IBD, represented by triangles. Dots indicate the known sample classification (healthy control, at y = 1 and blue; IBD, at y = 0 and red).

**Table 1 sensors-22-09246-t001:** Baseline characteristics.

	Controls	Inflammatory Bowel Disease		
	(*n* = 63)	Crohn’s disease (*n* = 24)	Ulcerative colitis (*n* = 39)	Total IBD (*n* = 63)
Sex, ♀ (*n*, %)	39 (61.9)	15 (62.5)	24 (61.5)	39 (61.5)
Age, mean ± SD	56.2 ± 11.1	34.2 ± 25.7	50.5 ± 17.6	44.3 ± 22.3
Smoking (*n*, %)				
Current	8 (12.7)	6 (25.0)	2 (5.1)	8 (12.7)
Past	22 (34.9)	8 (33.3)	14 (35.9)	22 (34.9)
Never	33 (52.4)	10 (41.7)	23 (59.0)	33 (52.4)
Disease activity (*n*, %)				
Quiescent	N.A.	5 (20.8)	17 (43.6)	22 (34.9)
Active	N.A.	19 (79.2)	22 (56.4)	41 (65.1)
Diet (*n*, %)				
None	53 (84.1)	18 (75.0)	33 (84.6)	51 (81.0)
Vegetarian	2 (3.2)	1 (4.2)	0 (0)	1 (1.6)
Gluten-free	3 (4.8)	2 (8.3)	3 (7.7)	5 (7.9)
Lactose-free	2 (3.2)	1 (4.2)	0 (0)	1 (1.6)
Other	5 (7.9)	2 (8.3)	3 (7.7)	5 (7.9)
Indication for endoscopy * (*n*, %)				
Positive FIT test	5 (7.9)	0 (0)	2 (5.1)	2 (3.2)
Rectal blood loss	8 (12.7)	3 (12.5)	3 (7.7)	6 (9.5)
Change in bowel habits	10 (15.9)	0 (0)	0 (0)	0 (0)
Surveillance †	14 (22.2)	0 (0)	20 (51.3)	20 (31.7)
Abdominal pain	13 (20.6)	3 (12.5)	0 (0)	3 (4.8)
Diarrhea	5 (7.9)	1 (4.2)	0 (0)	1 (1.6)
Family history of CRC	4 (6.3)	0 (0)	0 (0)	0 (0)
Follow-up after diverticulitis	2 (3.2)	0 (0)	0 (0)	0 (0)
Weight loss	3 (4.8)	0 (0)	0 (0)	0 (0)
Constipation	3 (4.8)	0 (0)	0 (0)	0 (0)
Anemia	1 (1.6)	0 (0)	0 (0)	0 (0)
Disease monitoring	N.A.	6 (25)	0 (0)	6 (9.5)
Suspected exacerbation	N.A.	10 (41.7)	12 (30.8)	22 (34.9)
Other **	N.A.	3 (12.5)	0 (0)	3 (4.8)

Abbreviations: *n*, number; SD, standard deviation; CD, Crohn’s Disease; UC, Ulcerative Colitis; IBD, Inflammatory Bowel Disease; NA, not applicable. * For seven controls and two CD patients, multiple indications were noted; for two UC patients, indication was insufficiently reported † Surveillance after CRC, for polyps and Lynch taken together ** other reasons for endoscopy in CD patients were medication adjustments (*n* = 1), monitoring pre-fistula surgery (*n* = 1) and monitoring after ileocecal resection (*n* = 1).

## Data Availability

Anonymized raw sensor output data as well as the statistical R script and output are openly accessible via the following link: https://doi.org/10.6084/m9.figshare.21547794.v1 (accessed on 18 November 2022).

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
