# Peer review of "Electronic Nose Sensor Drift Affects Diagnostic Reliability and Accuracy of Disease-Specific Algorithms"

_sensors, 2022, doi:10.3390/s22239246_

Round 1

Reviewer 1 Report

The authors observed the sensor drift effects on the Cyranose 320® E-Nose instrument. Based on the results, the authors claimed that a drift correction method was proposed. However, several critical issues must be addressed.

1) In the methods section, the authors mentiedn that a correction for sensor output was calculated 244 based on control samples using logistic regression. Only one sentence of the method was described. It is too simple without any details, equations, and procedures.

2) According to the result, only one single Figure.4 is too simple to convince the corrected performance.

3) Also, the quantitative error comparison should be given. The error bar, repeatability, and accuracy, as well as other results should be given.

4) The state-of-art technologies review or discussion was not suitable to appear at the end of the paper, which results in a very long but very odd Discussion part. It should be re-written in the Introduction part

5) Also, the references in such 'drift correction' discussion are very old. e.g. many of the refs[27-41] are ten yeas ago. I barely see any publications in 2021 and 2022.

6) Please cite the very new drift correction techniques published on authoritative journals e.g.

https://doi.org/10.1016/j.snb.2021.130986

https://doi.org/10.1016/j.snb.2022.131668

Author Response

To the editors and reviewers,

Thank you for the thorough review of the manuscript entitled: ‘Electronic Nose Sensor Drift affects Diagnostic Reliability and Accuracy of Disease specific Algorithms", and for the opportunity to write a point-to-point response to the reviewers. We have made multiple adjustments to the manuscript and believe our results have now become more comprehensible to the readers.

Reviewers’ comments

Reviewer 1:

1) In the methods section, the authors mentioned that a correction for sensor output was calculated 244 based on control samples using logistic regression. Only one sentence of the method was described. It is too simple without any details, equations, and procedures.

We agree with the reviewer this can be made clearer. We have now modified this in the Methods section:

“We fitted a regression model based on the control samples to correct for date of measurement for all samples, yielding a date-correction model for each sensor. The residuals of this model applied to the patient data then yield date-corrected measurements, used for subsequent analyses. ”

Should the reviewer believe that adding our R script and our raw data files would improve comprehensibility of our methods, we are of course willing to add this as supplementary file online in for example Figshare.

2) According to the result, only one single Figure.4 is too simple to convince the corrected performance.

We agree with the reviewer that only the data presented in Figure 4 is too little to convince any reader of the performance of fecal VOCs to detect inflammatory bowel disease (IBD). The high diagnostic accuracy of fecal volatile organic compound (VOC) analysis to discriminate IBD patients from controls has been shown in multiple previous studies. The main goal of this manuscript was, however, not to convince the reader on the diagnostic potential of VOC profiles for IBD detection, but to warn for the influence of sensor drift on the diagnostic potential of electronic Noses.

To us, the most important figures of this paper are Figure 2E and Figure 3. In figure 2, we have presented that ‘timing of measurement’, or in other words, sensor drift, affects sensor output more than any other variable (presence of bowel disease, previous bowel surgery, diet etc). Without correction for this, test accuracy is lost for IBD detection.
In figure 3, we present data distribution after correction for date of measurement. Even though all sensor data is corrected by using the change in resistance within the sensors when measuring control samples over time and not by using the change in resistance of blank samples run in between, it is possible to retrieve some levels of significance for 6 of 32 sensors to detect IBD. We have commented on this in the results section.

With these results, we do not aim to convince the readers of the performance of this test; we aim to present the effects of sensor drift and the possibility of improving diagnostic potential when correcting for this. 3) Also, the quantitative error comparison should be given. The error bar, repeatability, and accuracy, as well as other results should be given.

We have only applied leave-one-out cross-validation, so there is a single predicted probability available per patient. The sample size is too limited to warrant using double-loop cross-validation to assess replicability per sample. For this reason, it is not possible to produce error bars either. We have added accuracy, sensitivity and specificity in the text (just before figure 4). To extract more information of the current predictions, we have computed the Brier score, essentially the mean squared error associated with the predicted probabilities and have included boxplots of the prediction errors per group as supplementary figure.

4) The state-of-art technologies review or discussion was not suitable to appear at the end of the paper, which results in a very long but very odd Discussion part. It should be re-written in the Introduction part

We thank the reviewer for this useful comment. We agree with the reviewer on the fact that the discussion is too long. We have erased parts of the discussion section to improve comprehensibility of this paper and have added some of this to the introduction section.

5) Also, the references in such 'drift correction' discussion are very old. e.g. many of the refs[27-41] are ten yeas ago. I barely see any publications in 2021 and 2022.6) Please cite the very new drift correction techniques published on authoritative journals e.g.

https://doi.org/10.1016/j.snb.2021.130986

https://doi.org/10.1016/j.snb.2022.131668

Answer to both comment 5+6: we have re-written our discussion section and have included the references the reviewer has suggested.

Reviewer 2 Report

The research undertaken in this work is important and up-to-date, so the authors presented interesting research. I am only surprised that a significant signal drift was observed after just 2 days of measurement using the e-nose blowing procedure to maintain the baseline. Please comment on this. Below are my other comments:

line 60-62, which was the method of correction of the sensor signal drift. The authors did not describe it in the introduction, and this is the main goal of this research.

line 64-65, what exactly is an IBD biomarker? what are the chemicals?

line136-137, why is the room temperature set to 18 degrees? Higher temperature generates higher concentrations of volatile organic compounds. Please comment on this.

line 147-152, for the e-nose to work properly, after exposing the sensors to a given smell, return to the original point by passing air over the sensors. It is not stated how long a single analysis takes and how long it takes to blow the e-nose with air for the sensors to return to their original state.

line 218-219, what were the sensors? providing information that sensors 6, 19, 24 are not very correlated does not tell the reader anything.

line 244-246, it is not clearly described what the signal drift correction consisted of. On the other hand, it is quite strange that the drift was noticeable after 2 days of measurement. Please comment what could be the reason for this situation. And please describe in detail what is the correction of signal drift.

line 272-274, two sentences are to be removed

Author Response

To the editors and reviewers,

Thank you for the thorough review of the manuscript entitled: ‘Electronic Nose Sensor Drift affects Diagnostic Reliability and Accuracy of Disease specific Algorithms", and for the opportunity to write a point-to-point response to the reviewers. We have made multiple adjustments to the manuscript and believe our results have now become more comprehensible to the readers.

Reviewers’ comments

Reviewer 2:

The research undertaken in this work is important and up-to-date, so the authors presented interesting research. I am only surprised that a significant signal drift was observed after just 2 days of measurement using the e-nose blowing procedure to maintain the baseline. Please comment on this. Below are my other comments:

We thank the reviewer for their kind remarks on the importance of this topic.

line 60-62, which was the method of correction of the sensor signal drift. The authors did not describe it in the introduction, and this is the main goal of this research.

We agree with the reviewer that our method section should be made more clear. In the current study, we assessed sensor variation within the control group. As it is thought that healthy subjects have similar VOC profiles (with small variations due to medical, gender, BMI, smoking habits etc), we used this group for data normalization. We therefore assessed the mean relative resistance change per sensor per day for measurements within the control group, and corrected sensor output per sensor per day for IBD samples using this data. We have added this information to the methods section of this manuscript.

line 64-65, what exactly is an IBD biomarker? what are the chemicals?

Multiple studies have previously presented significant differences in VOC patterns between IBD patients and controls using a variety of techniques: eNose technology, Field Asymmetric Ion Mobility Spectrometry and Gas Chromatography Ion Mobility Spectrometry.

Most of these techniques focus on the entire spectrum of volatile organic compounds rather than on specific chemicals. Using all the volatile data, neural network analysis enables differentiation between patients with colonic diseases and controls, based on so called ‘smell prints’. Advantage of these techniques is they are easy to perform, so use in clinical practice can be done by clinicians without the need for specialized laboratory skilled personnel. In addition, measurements are fast and costs are relatively low compared to chemical-specific techniques.

line136-137, why is the room temperature set to 18 degrees? Higher temperature generates higher concentrations of volatile organic compounds. Please comment on this.

We agree with the reviewer that higher temperatures allow for the generation of higher VOC concentrations. The study pipeline for the Cyranose 320© has been validated using this current workflow specifically to improve usability in clinical settings. For example for the analysis of fecal samples by consultants during a consult at the outpatient clinic, which could spare more (unnecessary) invasive diagnostic procedures. As differences between IBD patients and controls have been presented using this (clinically relevant) work flow, we opted to maintain this operational procedure for the current study.

line 147-152, for the e-nose to work properly, after exposing the sensors to a given smell, return to the original point by passing air over the sensors. It is not stated how long a single analysis takes and how long it takes to blow the e-nose with air for the sensors to return to their original state.

We purged the sensors with air during 90 seconds after every measurement. This is the duration advised in the manual by the producers (Sensigent). Therefore, we believe this should be long enough for the residual VOCs to leave the sensors. In addition, baseline measurements with filtered air need to be performed every 5 to 6 measurements, and cross validation is performed within the apparatus, to avoid influence of residual VOCs on sensor outcome.

line 218-219, what were the sensors? providing information that sensors 6, 19, 24 are not very correlated does not tell the reader anything.

The sensors consist of a variety of nanocomposite polymers. Exact sensor characteristics are within intellectual property of Sensigent. As stated by the producer of this apparatus, each sensor has specific conducting characteristics, which result in unique changes in electrical resistance per sensor when different compounds pass the array.

Link: Cyranose 320 Select QA/QC Applications (sensigent.com)

The fact that most sensor outcomes are correlated in the current study means that the fecal pattern is not as unique for each sensor as suggested by the producers. We have presented that most sensors (apart from sensor 6, 19 and 24) have similar response to a singular sample.

We have added some thoughts on this outcome to the discussion section of the manuscript.

line 244-246, it is not clearly described what the signal drift correction consisted of. On the other hand, it is quite strange that the drift was noticeable after 2 days of measurement. Please comment what could be the reason for this situation. And please describe in detail what is the correction of signal drift.

In our study, the sensors were purged with filtered air after every measurement. This is air tapped from a controlled air container in the hospital. In addition, we used a strict operation procedure in which the samples were prepared by the same person, using the sample gloves, the same thawing manner and all samples were measured in the same room. As we kept all circumstances stable, we believe that only drift of sensors may have affected sensor resistance over days. Nevertheless, we agree with the reviewer that the extent of drift was surprisingly large in this study.

Even though sensor drift is a well-known disadvantage of eNose analysis, the extent to which this drift affects sensors over days has never been published in real-life cohort data. We do believe that raising awareness among researchers on the major effects sensor drift may have on diagnostic potential of diseases is very important. Should researched be unaware of this when performing clinical trials; samples may not be measurement in random order and this may results in false positive test characteristics caused by bias.

As described above, we used the control group for data normalization. We assessed the mean relative resistance change per sensor per day for measurements within the control group, and corrected sensor output per sensor per day for IBD samples using this data

line 272-274, two sentences are to be removed

Thank you for this comment. We have removed this remaining part of the template.

Round 2

Reviewer 2 Report

I accept most of the authors' responses to my remarks and comments. However, I believe that the explanation of the signal drift from individual sensors has not been properly explained, especially since the authors themselves are surprised by this phenomenon. Please have an additional discussion in Chapter 4.

Author Response

I accept most of the authors' responses to my remarks and comments. However, I believe that the explanation of the signal drift from individual sensors has not been properly explained, especially since the authors themselves are surprised by this phenomenon. Please have an additional discussion in Chapter 4.

As previously addressed, we do not have the explanation for the drift of individual sensors in this study. Though, sensor drift is a common phenomenon in chemical analytical studies. We have elaborated more on our hypothesis for the origin of the short term sensor drift in the current study. This has been added to the discussion section of the manuscript.